# Spray-Dried Animal Plasma as a Multifaceted Ingredient in Pet Food

**DOI:** 10.3390/ani13111773

**Published:** 2023-05-26

**Authors:** Ricardo Souza Vasconcellos, Lucas Ben Fiuza Henríquez, Patrick dos Santos Lourenço

**Affiliations:** Department of Animal Science, State University of Maringá, Maringá 87020-900, Brazil; pg55187@uem.br (L.B.F.H.); ra107443@uem.br (P.d.S.L.)

**Keywords:** animal nutrition, dogs, cats, functionality, binder, emulsifier, immunomodulatory, prebiotic, cognition

## Abstract

**Simple Summary:**

In this review, the nutritional value and functional properties of spray-dried animal plasma (SDAP) for use in pet food are discussed. Although SDAP is a co-product of meat production for human consumption, this ingredient has a high nutritional standard, technological properties for wet and dry pet food production, and its bioactive components have high potential for use with a focus on intestinal health, immunity, neuroprotection, and control of “inflammageing” in animals. These properties are discussed in this literature review.

**Abstract:**

Plasma is a co-product from pork and beef obtained during the processing of animals for human consumption. The spray-drying process maintains the solubility of spray-dried animal plasma (SDAP) and its nutritional and functional properties, making this ingredient multifunctional in human and animal nutrition. In pet food, SDAP has been used in the production of wet foods (pates and chunks in gravy) as an emulsifying and binding agent, with the potential to replace hydrocolloids partially or totally, which have some negative implications for digestibility, fecal quality, and intestinal inflammation. From a nutritional point of view, SDAP has high digestibility and an amino acid profile compatible with high-quality ingredients, such as powdered eggs. Studies in companion animals, especially in cats, have shown that SDAP is an ingredient with high palatability. Despite the immunomodulatory, anti-inflammatory, prebiotic, and neuroprotective properties demonstrated in some animal models, there are still few publications demonstrating these effects in dogs and cats, which limits its use as a functional ingredient for these species. In this review, the potential use of SDAP in pet food, aspects related to the sustainability of this ingredient, and opportunities for studies in companion animals are discussed.

## 1. Introduction

In the past decade, increased demand for high-quality raw materials, supply chain disruptions, and demand for ingredients with a low environmental impact have created many challenges in the animal feed and pet food industry [1]. The humanizing desire for high-quality food for dogs and cats along with the demand for highly digestible, palatable, and biologically valuable ingredients has created challenges and opportunities. There is an increased interest in protein ingredients that have functional properties, meet nutritional requirements, optimize health, and reduce the environmental impact. Some functional animal and plant-origin ingredients have the potential to meet these demands and spray-dried animal plasma (SDAP) is one of them.

Plasma is a co-product from pork and beef obtained during the processing of animals for human consumption. Anticoagulants are added to the blood during collection. Then, the blood is centrifuged to separate the plasma and cellular fractions (red blood cells and platelets), and these fractions are subsequently spray-dried into a powder form (Figure 1). This processing system maintains the solubility and functionality of the ingredient when added to dry or wet food for dogs and cats.

SDAP is used in both human and animal food. In pet food, it can be used for several purposes. For wet food with a high moisture and fat content that has the possibility of particle segregation, the main technological application of SDAP is as an emulsifying and binding agent (“binder”) to improve water retention, texture, juiciness, and homogenization of the final product [2,3,4]. In dry and extrusion-processed pet food, the technological properties of SDAP have not yet been widely explored, although emulsifying agents and binders are commonly used to improve the kibble characteristics.

The inherent solubility and functional properties of plasma are maintained after manufacturing. Plasma as a fraction of animal blood is a very nutritionally rich ingredient with high concentrations of essential amino acids and bioactive components that confer its biological functionality [5,6]. The high amino acid concentration, immunoglobulins, bioactive peptides, growth factors, enzymes, and metalloproteins provide immunomodulatory [6,7,8], prebiotic [5,9,10], anti-inflammatory [9,11] and neuroprotective properties [5,12,13]. Figure 2 shows a summary of the main applications of this ingredient in dog and cat food.

Considering the pet food processing, functional properties, and nutritional advantages of SDAP, this review aims to discuss the results of available studies with this ingredient, its main applications currently used by the industry, and perspectives for innovative uses in dog and cat nutrition.

The following areas will be covered: sustainability of SDAP, wet pet food processing, nutritional value for dogs and cats, immunological effects, the use in aging animals, limitations on the use in pet diets, future options applying SDAP in pet food, and conclusions and perspectives.

## 2. SDAP as a Sustainable Ingredient

Historically, animal blood from the meat industry was considered a waste product with the potential to cause environmental pollution. Therefore, it was processed for use in animal feed, and this destination was considered the most viable from an economic and environmental point of view [14]. Higher-value animal blood co-products obtained from industrial processes to produce beef or pork for human consumption, such as SDAP, have since been developed. The process of using waste from the production of food for humans and its use in other stages of the production chain is known as “Industrial Ecology” [15] and, in the case of slaughterhouse waste, “Rendering or Animal Recycling” [14]. Animal recycling contributes significantly to the reduction in the emission of greenhouse gases if the destination of these co-products were different, such as landfills or composting [14].

Life Cycle Assessment (LCA) is a way to quantify the environmental impacts of production and use of products and has been applied as a tool in choosing ingredients in animal formulations, aiming to minimize the impacts of formulations on animal production. For this reason, industrial co-products of animal and vegetable origin that are not used for human food benefit the circular economy, which promotes the continued use of parts of a primary resource, successively minimizing the waste generated while at the same time giving purpose to the co-products generated in the different stages of the production chain. Especially, meat co-products minimize environmental impacts and have been the focus of research to enhance their nutritional value [16,17].

Pet food uses many co-products of plant and animal origin that are not consumed by humans. Although the volume of pet food produced globally represents less than 10% of the global production of animal feed, this market uses more than 30% of all co-products generated in the process of animal recycling and therefore, has an important role in the current use of industrial co-products. In the case of co-products from the rendering process, LCA recommends that the allocation of their impacts be performed economically [18,19] since it would not be correct to attribute similar impacts to the products generated from rendering, which precisely has the role of not disposing meat production waste to the environment.

Ingredients such as beef tallow, chicken fat, and animal-based meal, including SDAP, have an important role in minimizing the environmental impacts caused by the production of food for human consumption in the circular economy.

## 3. Wet Pet Food

The protein content ranges from 70% to 80% of the final spray-dried plasma ingredient, depending on the membranes used for plasma concentration before drying. The drying stage uses reverse osmosis to remove water for the product with 70% protein or ultra or nanofiltration that, in addition to water, removes some salts for the product with 80% protein. SDAP is mainly composed of albumin, immunoglobulin G (IgG), and coagulation proteins [20]. Because the spray-drying process preserves the quality of plasma components and maintains their solubility [21], SDAP has important functional properties in wet food processing to produce pate or chunks in gravy and can be used as an emulsifying and binding agent, having a role like hydrocolloids [20].

Hydrocolloids represent a diverse group of readily dispersible, fully, or partially water-soluble, long-chain polymers that increase in volume in water. They change the physical properties of the environment by forming gels, thickening, emulsifying, recoating, and stabilizing food components [22]. Although plasma is not included in the hydrocolloid group, which is essentially composed of polysaccharides and collagen, SDAP has very similar properties and is commonly used in wet foods for this purpose. For this review, a market investigation by one of the largest e-commerce retailers in Brazil (www.petlove.com, accessed on 3 February 2023) was performed; among 25 brands of wet pet foods, 44% of these wet pet foods declare SDAP in the composition, probably as an emulsifying or binding agent. Other typical hydrocolloid agents were declared in pet food that did not include SDAP, such as xanthan gum, guar gum, carrageenan gum, collagen, and modified starch. Other agents commonly used for this purpose are wheat gluten, soy protein, and whey protein.

Wet pet foods are mainly formulated with co-products from slaughterhouses that have high protein, lipid, and moisture content (70–85%). Gelling agents and emulsifiers are used to avoid particle segregation and improve the texture and homogenization [23].

In a study comparing the technological properties of binders commonly used in wet food to form chunks in gravy or loaves, the inclusion of 1.5% or 2.5% SDAP increased the hardness and reduced water loss when compared to wheat gluten [4]. Similarly, in another study comparing the inclusion of 2% SDAP with 2%, 4%, or 6% wheat gluten, there was a 2.5-fold increase in chunk hardness for the 2% SDAP, and juiciness was improved by approximately 20% because of the increased absorption of water from the gravy in contact with the chunk [24]. Juiciness was defined in that study as the amount of gravy that was absorbed by the chunks after cooking and corresponding to the change in weight of chunks after cooking and storage for two weeks. Therefore, juiciness can be described as the ability of meat to release juice (defined as the quantity of water preserved after cooking) when a pet chews it. The sensation of juiciness happens in two ways: the first corresponds to the quantity of juice that is released into the mouth when the meat is first chewed, and the secondary perception of juiciness is due to salivary flow stimulated by the presence of fat in the mouth. Juiciness or succulence is well correlated with improved palatability and enjoyable eating for humans and our pets.

Figure 3 demonstrates the emulsification properties of SDAP in pate products.

SDAP confers toughness to the pate due to its high-water retention capacity when included at 20% in the recipe as compared to 20% wheat gluten (WG). This is shown in Figure 4.

No studies were found in the literature that directly compared the gelling properties of SDAP and hydrocolloids. Hydrocolloids are considered additives, while SDAP is a high-protein ingredient for pet food that provides technological and nutritional properties, and it is used mainly because of these characteristics. Among the typical hydrocolloids, carrageenan gum has a high gelling and emulsifying capacity [23] and together with other hydrocolloids, such as xanthan gum, guar gum, or locust bean gum, are the most used in pet food.

Although various gums have important technological actions, there are some undesirable effects associated with their use [25]. Dogs that were fed diets containing 0.4% guar gum had reduced protein digestibility and stool quality [26]. Similar effects were also reported with a mixture of guar gum and carrageenan, although the authors also reported intestinal fermentative benefits with these additives [27]. Another undesirable effect of the use of hydrocolloids is their ability to induce intestinal inflammation and gastric ulcers as described in rats, mice, rabbits, and guinea pigs after ingesting carrageenan gum or carboxymethyl cellulose [28,29].

On the other hand, faster healing of gastric ulcers was observed in pigs after ingesting plasma proteins in their drinking water when compared to the control group that did not receive the addition of these proteins in the water [30]. In addition, performance horses that consumed a supplement with plasma-based proteins prevented the development of stress-induced gastric ulcers [31]. However, no studies have evaluated the effects of plasma proteins on ulcers in dogs and cats. SDAP is a highly digestible ingredient with immunomodulatory and anti-inflammatory properties that could be used at higher levels to partially reduce the inclusion level of hydrocolloids to avoid undesirable effects while maintaining the desirable technological characteristics in the pet food product.

## 4. Nutritional Value for Dogs and Cats

The main nutritional attribute of SDAP is its high protein concentration with an appropriate composition of essential amino acids compared to other ingredients used in pet food. However, due to the differences in protein concentrations between ingredients, the amino acid score (AAS) calculation represents an adequate comparative measure between protein sources [32]. The AAS is calculated by the amino acid concentration within the food source, divided by the recommended intake of that amino acid from reference tables for the species. In the case of dogs and cats, currently, the most used tables are from the European Pet Food Industry Federation (FEDIAF) and the Association of American Food Control Officials (AAFCO).

Compared to other protein ingredients, SDAP has the highest total AAS for lysine, tryptophan, and threonine, which, together with methionine, are the main limiting amino acids in diets for dogs and cats [33]. The superior amino acid profile of SDAP is also accompanied by the higher digestibility of this ingredient in dry [34] and wet [35] foods, which improves the biological value of the protein compared to conventional sources. Table 1 shows a comparison between SDAP and the main protein sources used in pet food [18,36].

The AAS of SDAP is adequate and superior to other ingredients commonly used in pet food but is relatively low in methionine (Figure 5).

Plasma protein is composed mainly of albumin, globulins, and other components normally found in the blood. Because of this composition, the main nutritionally focused application is to meet biological needs for amino acids. The nutritional value of a protein source is determined primarily by its digestibility and its amino acid composition concerning the species’ requirement (AAS), so the latter directly influences the biological value [38].

As shown in Table 1, SDAP has an amino acid profile comparable to high-quality protein sources such as egg powder, except for the relatively lower levels of methionine in SDAP. There have been a few studies evaluating SDAP as a protein ingredient in diets for dogs and cats. In three studies, dogs that were fed diets with up to 3% SDAP had increased apparent dry matter (DM) digestibility in all experiments and had increased crude protein digestibility in two of the studies, with no changes in stool quality [34]. Similar results were found using cat diets containing 3% SDAP or 3% wheat gluten as binders [35]. Cats that were fed diets with SDAP had higher DM digestibility without any significant changes in protein digestibility, and there were significant reductions in the volume of feces excreted by the animals.

While few studies have evaluated the digestibility coefficients of SDAP in companion animals, this ingredient has been studied more extensively in pigs and poultry to minimize the use of antibiotics and improve immunity indicators, due to its potential benefits on the digestive system. The digestibility of SDAP by poultry and swine is equivalent to other highly digestible ingredients such as egg powder [39], brewer’s yeast extract [29,40], whey protein concentrate [41], and hydrolyzed fish meal [41]. Despite the similarity with other highly digestible ingredients, SDAP has consistently shown additional benefits in weight gain, feed conversion, and improved disease resistance when included in diets for weaned pigs [42].

The biological value of ingredients is important because protein sources with high biological value can be included in lower concentrations and can reduce formulation costs. A low biological value for wheat gluten was observed, resulting in weight loss in puppies that were fed low-protein diets when compared to other sources such as hydrolyzed casein and soy protein isolate [38].

A widely used measure of protein quality in humans is the Protein Digestibility Corrected Amino Acid Score (PDCAAS). This measure is obtained by multiplying the amino acid score (AAS) by the apparent protein digestibility of a given source [32]. Using the data in Table 1 and the reported digestibility of 90% for both SDAP and WG [38,39], the PDCAAS for lysine, methionine, and tryptophan are very low for WG, which contributes to the low biological value of this ingredient, while SDAP presents lower values only for methionine. Among the protein sources compared in Figure 6, the PDCAAS of SDAP is higher for the main amino acids than the other ingredients, including egg powder, with the exception that methionine and cystine are lower in SDAP.

## 5. Immunological Effects of SDAP

Dietary SDAP also has immunological properties in the digestive system, but these effects have not been determined for dogs and cats. In swine and rodent models, some mechanisms of action of SDAP in intestinal protection are proposed in Table 2.

Among the benefits to the digestive tract with the use of SDAP, its prebiotic and modulating effects on local gut immunity have been studied. In a study comparing the effects of 0% or 8% SDAP in diets fed to mice on intestinal microbiome populations [9], mice that were fed SDAP had a significant increase in the count of microorganisms of the Phylum Firmicutes that is generally associated with the production of short-chain fatty acids (SCFA), which regulate the growth of pathogenic microorganisms and improve gut immunity. Among the genera that showed the highest increase were *Lactobacillus* spp. and *Blautia* spp. The changes in the intestinal microbiota for mice that were fed SDAP were associated with an increased expression of mucosa interleukin 10 (IL-10), transforming growth factor beta (TGF-β), mucin 2 (MUC-2), and trefoil factor 3 (TFF3), all of which increase immune tolerance of the intestinal mucosa to minimize the risk of inflammation. Similar results of SDAP modulating the intestinal microbiota have been observed in pigs and even in fish [45,46].

Part of the intestinal protector effects associated with the consumption of SDAP are related to its high concentration of immunoglobulins, especially IgG, which confers a direct effect against pathogenic microorganisms and prevents lesions in the intestinal mucosa. Mice genetically predisposed to inflammatory bowel disease were fed diets containing 2% bovine serum immunoglobulin isolate (BSI) [10] or 8% SDAP [5]. Both studies showed that BSI and SDAP prevented the increase of pro-inflammatory mucosal cytokines and chemokines including IL-2, IL-6, IL-17, MIP-1β, and MCP-1. Furthermore, BSI increased the expression of anti-inflammatory cytokines TGF-1β, while SDAP increased IL-10. Thus, immunomodulatory components in SDAP and BSI play an important role in protecting and improving intestinal immune tolerance.

The immune modulatory effects related to SDAP supplementation are not limited to the local intestinal mucosa. It was already observed that dietary SDAP reduced acute pulmonary inflammation induced by lipopolysaccharide inhalation in mice [47] and improved pregnancy rates and favorably altered the uterine mucosa cytokine profile in transport-stressed mice [48].

An important aspect to consider when using bioactive compounds in animal nutrition is their viability along the digestive tract. Studies have verified in dogs and cats that 7.6% and 4.9%, respectively, of the porcine IgG ingredients consumed remain viable in feces, demonstrating that IgG can have activity throughout the digestive tract [49]. However, the porcine IgG ingredients used in these studies were applied after the kibbles were extruded because it was unknown if the extrusion process could inactivate their functions.

Plasma proteins in immunologically challenged animals show positive effects on the growth rate, feed intake, and general health conditions when compared to animals not receiving these proteins. Weaning stress can reduce feed consumption, resulting in a compromised intestinal barrier and predisposing pigs to diarrhea and infections [50]. A recent meta-analysis [6] reported that increasing the concentrations of SDAP by up to 10% in the diet for weaned pigs significantly increased feed intake, showing that SDAP was palatable and minimized the negative impacts of postweaning stress on pig performance. The mechanisms of action and the effective concentration of SDAP in the diet of various animal species still need to be better defined (Table 3).

However, the SDAP effects on animal performance improvement are not only attributed to the increased feed intake but also to the presence of immunoglobulins, growth factors, and other proteins present in the plasma, which have a positive effect against important intestinal pathogens such as *Salmonella typhimurium*, *Salmonella enteritidis*, *Escherichia coli*, *Staphylococcus* spp. and their enterotoxins [47,85,86]. The intestinal and immune-related benefits of SDAP seem to be mostly related to the activity of immunoglobulins and/or bioactive peptides produced during the digestion of SDAP in the gastrointestinal tract. Other authors have suggested a prebiotic effect of SDAP on promoting the growth of beneficial intestinal microbiota. This may help the control of pathogenic microbial populations in the gut and therefore reduce antigen stimulation and the subsequent production of pro-inflammatory cytokines and chemokines such as IFN-γ, TNF-α, IL-1, IL-6, and LTB-4. These compounds are associated with the activation of gut-associated lymphoid tissue (GALT), which promote mucosal lesions and predispose the animals to microbial translocation [9,58].

The effects of SDAP on increasing intestinal immune tolerance seem to be primarily related to an increased production of inflammation-regulating cytokines and chemokines, such as IL-10, TGF-β, β-defensin, iNOS, integrins, and other adhesion molecules [86]. However, microbiota interactions can also modulate gut tolerance (Figure 7).

Inflammatory bowel disease (IBD) in dogs is the most common cause of intestinal disease and may be associated with parasitosis, food allergies, idiopathic inflammation, or a previously existing disease. The primary changes in dogs with IBD are increased intestinal permeability, increased dysbiosis index (Figure 8), increased cellular infiltrates of mononuclear cells, and increased pro-inflammatory cytokines (IL-2, IL-6, TNF-α, IFN-γ, and IL-1β) and C-reactive protein [88]. Although SDAP supplementation has not been studied in dogs with IBD, many of its attributes could help alleviate clinical conditions. Supplementation with BSI reduced irritable bowel syndrome in humans [89,90].

According to a systematic literature review [91] that included the results of 11 publications about animals that were immunologically challenged, a consistent effect of plasma protein-derived ingredient supplementation on daily weight gain, feed intake, and feed conversion was reported, along with a significant reduction in pro-inflammatory cytokines such as IL-6, TNF-α, and IL-1β. The authors suggested studies using humans with enteropathies were needed given the consistent findings in challenged animals. Similar results of dietary SDAP on pig performance were described in a systematic review [92]. However, in this review, only results from experiments without intentional challenges to the animals were used but improved physiological conditions in animals supplemented with SDAP were reported.

Inflammation associated with IBD may be caused by an alteration in the balance between Treg and proinflammatory-activated Th cells [71,88]. SDAP supplementation reduced the ratio between activated Th lymphocytes and Treg lymphocytes, indicating that SDAP restores the balance between these lymphocyte populations [11]. A similar response observed in mice genetically predisposed to IBD has been observed in a Staphylococcal enterotoxin B model of mild intestinal inflammation [51] and under acute lung inflammation induced by LPS [47].

## 6. SDAP in Aging Animals

The aging process in dogs and cats is very similar to humans [93]. General changes in body function occur because of reduced adaptive capacity, which in part results from a process known as “inflammageing”. This is characterized by mild to moderate immunological stimulation, which culminates in oxidative, inflammatory, and degenerative modifications in some organ systems [94]. Among all of them, the changes in the nervous system seem to be more evident because they are more easily perceived, since the animals show deficits in locomotion, sense organs, learning ability, changes in the sleep–wake cycle, and even in learned functions, such as the habit of defecating and urinating in inappropriate places. Such changes are part of a syndrome known as Cognitive Dysfunction, which, from a pathophysiological standpoint, is very similar to Alzheimer’s disease in humans [95]. Currently, the decline in cognitive functions has been partially associated with modifications in the gut microbiota [96], and for this reason, studies on the brain–intestine interaction (“gut-brain axis”) have gained relevance in the search for nutritional strategies to minimize the effects of aging in animals.

Based on this principle, the association between the low-level inflammation and the decline of cognitive functions in mice predisposed to early aging (SAMP8) and how supplementation with 8% SDAP was able to minimize the impacts of aging were studied [12]. The authors showed a significant improvement in the cognitive functions of animals supplemented with SDAP compared to the non-supplemented ones (negative control group) using short- and long-term memory tests. These effects were partially attributed to the reduction of pro-inflammatory (IL-6 and NF-κβ) and oxidative (hydrogen peroxide concentration) markers, increased expression of inflammation-modulating cytokine (IL-10), and increased adhesion molecules at the blood–brain barrier (e-cadherin and ZO-1), which reduce capillary permeability in the central nervous system, protecting neurons from potentially toxic compounds that can induce neuronal degeneration.

In addition, the neuronal protective effects of SDAP were attributed to its effects on the gut in two other publications [5,83] by this same research group, which demonstrated that aging in mice induces mild-grade inflammatory changes and that senescence promotes an increase in populations of potentially pathogenic microorganisms, with a reduction in microorganisms with probiotic potential (*Lactobacillus hayakitensis* and *Blautia hansenii*) that promote protection against infections by *Clostridium* spp. In this study [83], supplementation of SDAP reversed such deleterious changes in the aging animals, promoting an effective response to an intestinal challenge.

Two of the main neuronal changes identified histopathologically in Alzheimer’s disease are the presence of the amyloid precursor protein (APP) and neurofibrillary tangles. In SAMP8 mice, the SDAP supplementation reduced the expression of proteins related to these two Alzheimer’s disease-related structures [13].

Despite the lack of studies using aging dogs and cats supplemented with SDAP, the results obtained with aging mice suggest important benefits for improving quality of life by minimizing the effects of inflammageing and preserving cognitive functions.

## 7. Limitations on the Use of SDAP in Pet Food Diets

There are some factors that can limit the use of SDAP in pet food. First, SDAP comes from an undervalued raw material (blood), but the process to separate the plasma from the red cells and the dehydration by spray-drying technology to maintain the high solubility of this ingredient increase its cost. So, pet food producers need to account for it in the cost of their final recipe, which could impact the inclusion levels in the product. Second, when formulating SDAP into diets or supplements, the nutrient profile including amino acid and minerals should be accounted for during formulation to meet desired nutritional requirements for complete food and supplements. Third, researchers still need to investigate whether SDAP should be applied before or after extrusion for optimal results [34]. When applying externally, methods of application such as tumbling or vacuum coating can impact effectiveness of application and limit the level applied. When blended internally before extrusion, further research is needed to understand the impact on SDAP biological activity, because only one digestibility study has evaluated the internal versus external application of SDAP [34].

## 8. Future Options Applying SDAP in Pet Food

Considering the multiple functions of SDAP presented in this review, it is important to further investigate its effects and concentrations in dogs and cats, due to its high potential for applications, especially in dry pet food, treats, and supplements. The use of SDAP in dry pet food opens new opportunities for pet food producers to develop a diversity of new concepts, such as the use of SDAP to improve cognitive functions and mobility in diets oriented to senior pets. Additionally, there is an opportunity to use SDAP in veterinary diets for pets with gastrointestinal disorders, especially for companion animals presenting IBD or other inflammatory disorders. Furthermore, due to the prebiotic effect of SDAP, its use in puppies and kittens can help develop a beneficial microbiota in early life stages to help pets develop a strong, healthy gastrointestinal tract. In addition, the binding properties of SDAP may help the development of dry pet food recipes with high fresh meat inclusion, while maintaining the physical properties of dry kibbles.

## 9. Conclusions and Perspectives

The nutritional composition of SDAP along with its bioactive components make this animal co-product an ingredient with high digestibility and biological value that also has prebiotic, immunomodulatory, and neuroprotective properties, with multiple potential new applications for its use in wet and dry pet food products. The technological benefits of SDAP as an emulsifier and binder in wet food are well demonstrated, and its use by the industry is widely recognized.

In addition to the above-mentioned properties which make SDAP a multifaceted ingredient, it also contributes to a reduced environmental impact when compared to other ingredients, even those of plant origin, since it is produced from industrial waste from human food production and is a sustainable ingredient.

Although all the benefits of SDAP have been demonstrated for dogs and cats, it is necessary to conduct more studies with a dose–response for SDAP, because publications about wet and dry pet foods with a technological and nutritional focus were evaluated with formula inclusions of up to 2%, while studies focused on its functional properties used 8%. Thus, there is a gap between the formula levels used that needs further study to determine the optimum dietary concentration of SDAP in various applications for pet food products.

## Figures and Tables

**Figure 1 animals-13-01773-f001:**
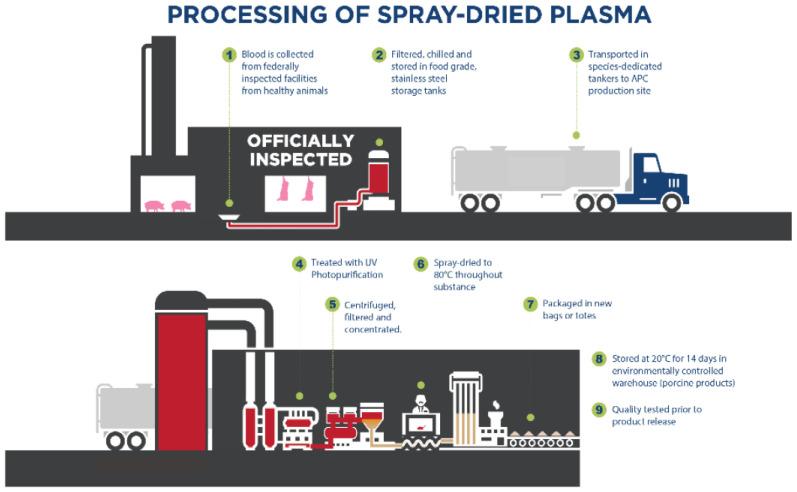
Production diagram of spray-dried animal plasma (SDAP), courtesy of APC LLC, Ankeny, IA.

**Figure 2 animals-13-01773-f002:**
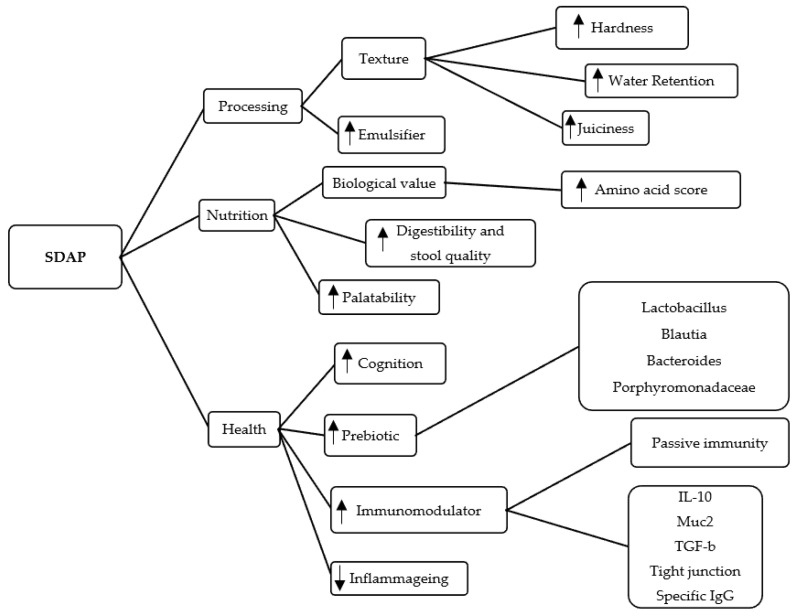
Flow chart of the main technological (processing), nutritional, and functional (health) properties of spray-dried animal plasma (SDAP) for use in pet food. Image from the author.

**Figure 3 animals-13-01773-f003:**
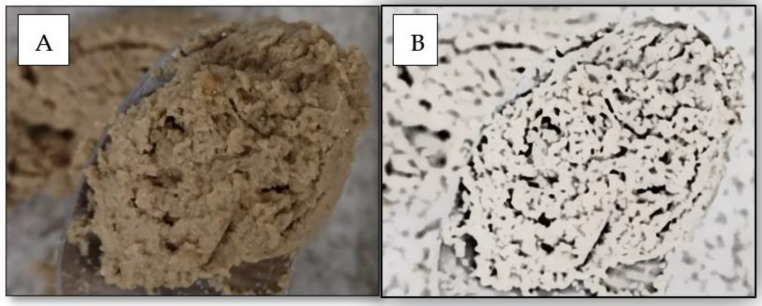
Color (**A**) and black and white (**B**) images of cat pate showing the air cells in the pate processed with SDAP due to its emulsifying properties. Images from the author.

**Figure 4 animals-13-01773-f004:**
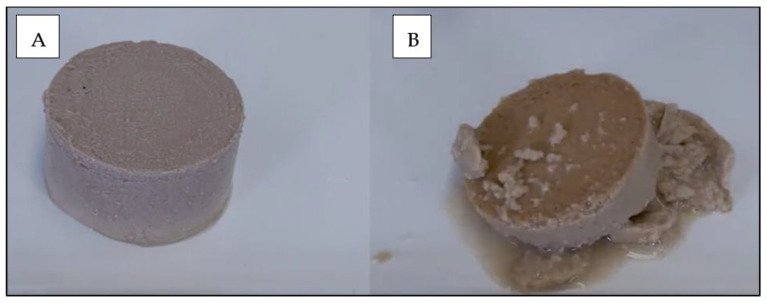
Picture of a pate produced with 20% SDAP (**A**) or 20% wheat gluten (**B**). The image shows that both can give texture and consistency to the food, but plasma has a greater capacity to retain water at the same concentration. Courtesy of APC LLC, Ankeny, IA, USA.

**Figure 5 animals-13-01773-f005:**
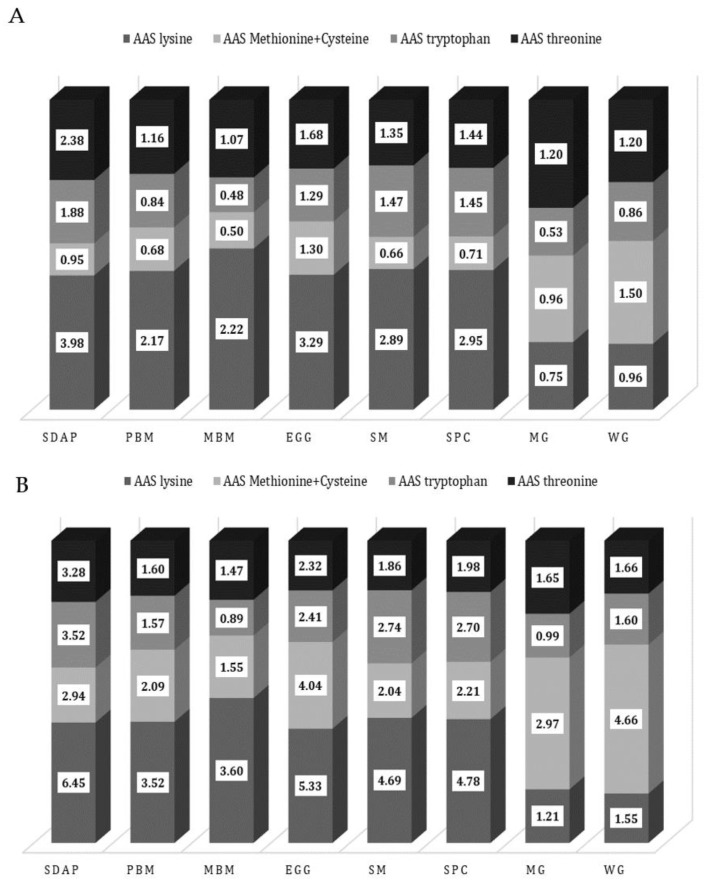
The amino acid score for dogs (**A**) and cats (**B**) for the main amino acids in commonly used ingredients for pet food. SDAP—spray-dried porcine plasma; PBM—poultry by-product meal; MBM—meat and bone meal; EGG—egg powder; SM—soybean meal; SPC—soy protein concentrate; MG—maize gluten feed; WG—wheat gluten. An amino acid score (AAS) is calculated by using the equation recommended by FAO [37], from the amino acid composition provided in Table 1.

**Figure 6 animals-13-01773-f006:**
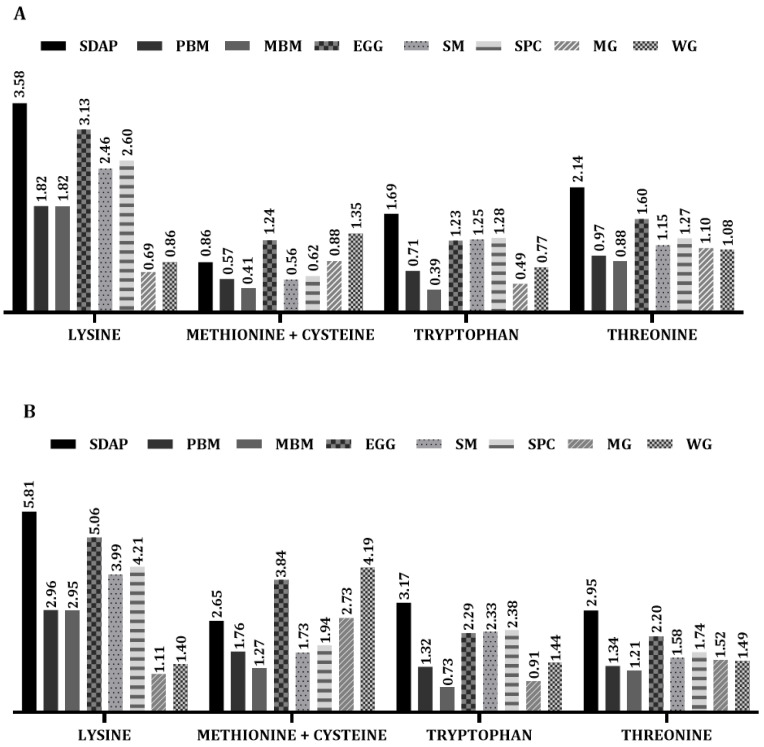
Protein Digestibility Corrected Amino Acid Score (PDCAAS) of different ingredients used in pet food for dogs (**A**) or cats (**B**). SDAP—spray-dried porcine plasma; PBM—poultry by-product meal; MBM—meat and bone meal; EGG—egg powder; SM—soybean meal; SPC—soy protein concentrate; MG—maize gluten feed; WG—wheat gluten. PDCAAS was calculated according to Schaafsma [43] from the ingredient composition and nutritional recommendation provided in Table 1.

**Figure 7 animals-13-01773-f007:**
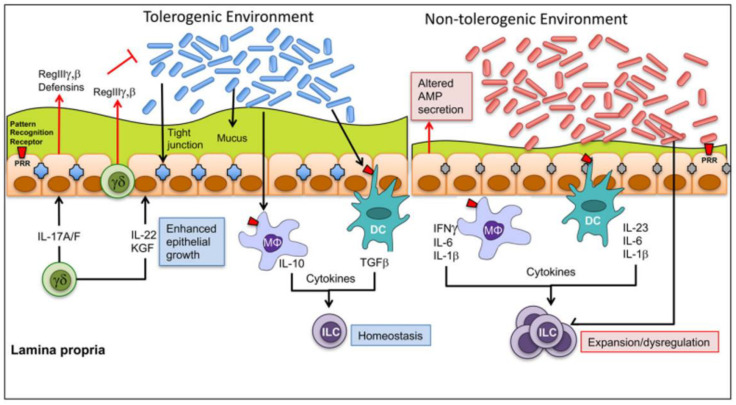
Microbiota interactions that can modulate gut tolerance. A gut tolerogenic environment is maintained by a tolerogenic gut microbiota, which leads to the formation of a healthy mucus coat, proper tight junction function, and a balance between epithelial and innate immune cells. Tolerogenic homeostasis is enhanced by the secretion of IL-10 and growth factor (TGF-β). In a non-tolerogenic environment, dysbiosis can favor the production of inflammatory cytokines, such as interferon IFN-γ, IL-6, IL-1b, and IL-23. Figure extracted with permission from [87].

**Figure 8 animals-13-01773-f008:**
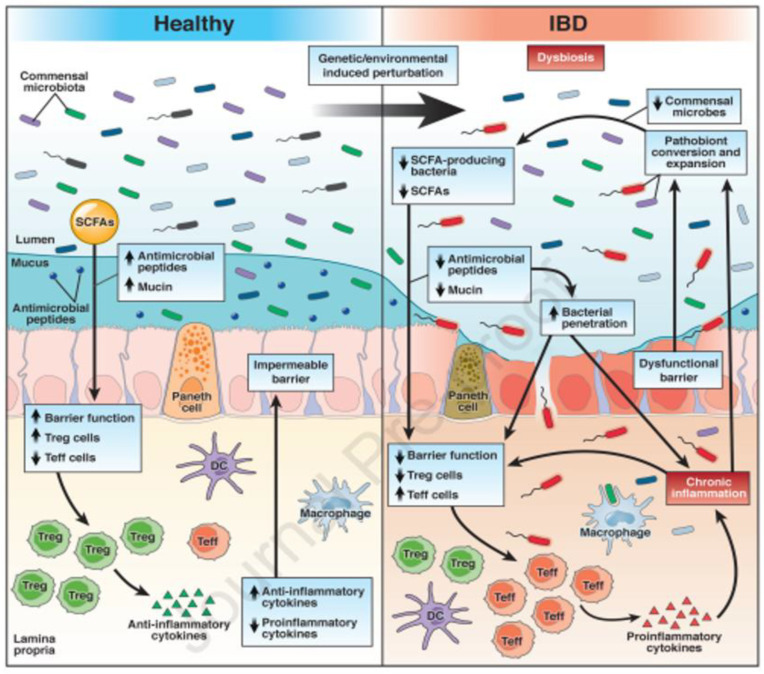
Gut indicators in individuals with a healthy gut environment or with dysbiosis, which promotes inflammation, with epithelium damage, microbial translocation, and immune intolerance. Many of the actions of SDAP contribute to a healthier gut environment, such as an increase in anti-inflammatory cytokines and a reduction in pro-inflammatory ones, a reduction in mucosal permeability, and balancing the gut microbiota. Figure extracted with permission from [51].

**Table 1 animals-13-01773-t001:** Protein content and amino acid composition of some protein sources, and the FEDIAF ^1^ recommendations for adult dogs and cats.

Item	FEDIAF ^1^ Dogs	FEDIAF ^1^ Cats	SDAP	PBM	MBM	EGG	SM	SPC	MG	WG
Crude Protein, %	21.0	33.3	78.0	65.0	44.0	47.2	46.5	63.1	61.1	79.8
Arginine, %	0.60	1.30	4.70	3.90	3.20	2.84	3.35	5.21	1.96	3.65
Histidine, %	0.27	0.35	2.80	1.07	0.67	1.12	1.21	1.72	1.28	1.95
Isoleucine, %	0.53	0.57	2.90	2.07	1.06	2.58	2.29	3.00	2.54	4.24
Leucine, %	0.95	1.36	7.80	3.89	2.29	4.05	3.56	5.07	10.6	7.29
Lysine, %	0.46	0.45	6.80	3.09	2.14	3.40	2.95	4.07	1.00	1.67
Methionine, %	0.46	0.23	0.60	1.06	0.56	1.48	0.61	0.92	1.38	1.75
Methionine + Cysteine, %	0.88	0.45	3.10	1.84	0.92	2.58	1.28	1.88	2.45	5.02
Phenylalanine, %	0.63	0.53	4.60	2.24	1.29	2.52	2.42	3.37	3.93	3.28
Phenylalanine + Tyrosine, %	1.03	2.04	8.20	3.71	2.05	4.45	3.81	5.73	7.16	6.56
Threonine, %	0.60	0.69	5.30	2.16	1.34	2.27	1.79	2.59	2.09	2.74
Tryptophan, %	0.20	0.17	1.40	0.52	0.20	0.58	0.65	0.87	0.31	0.65
Valine, %	0.68	0.68	5.30	2.67	1.62	2.89	2.14	3.16	2.86	4.05

^1^ FEDIAF [30]—Recommended nutrient levels for adult dogs (95 kcal/kg^0.75^) and cats (75 kcal/kg^0.67^), in units per 100 g of dry matter (DM); SDAP—spray-dried animal plasma; PBM—poultry by-product meal; MBM—meat and bone meal; EGG—egg powder; SM—soybean meal; SPC—soy protein concentrate; MG—maize gluten feed; WG—wheat gluten.

**Table 2 animals-13-01773-t002:** Major immunological modifications promoted by SDAP in the gut and effects on the intestinal mucosal barrier [44].

**Mitigation of Innate and Acquired Immune Response**
Reduction in activated lymphocytes and neutrophils Reduction in intestinal TNF-α and IL-1β expression Increased expression of IL-10 and TGF-β Reduction in Th17/Treg ratio Reduced expression of VCAM-1 and ICAM-1 A direct effect of IgG from SDAP in the elimination of microorganisms
**Effects Observed Due to Immunological Modifications**
Reduction in local inflammation Elimination of pathogens Modulation of the intestinal microbiota Improvement in mucosal integrity

**Table 3 animals-13-01773-t003:** Studied effects of SDAP in different species of animals, including pets, that were fed SDAP in their diet.

Known Effect of SDAP in Different Animal Species	References Supporting These Effects
SDAP supports and helps to maintain:	
− intestinal immune function	[51] ^2^, [52] ^2^, [53] ^2^, [54] ^3^, [55] ^3^, [56] ^3^;
− respiratory immune function	[47] ^2^, [57] ^2^, [58] ^2^, [59] ^3^, [60] ^3^;
− reproductive performance	[48] ^2^, [61] ^2^, [62] ^2^, [63] ^3^, [64] ^3^, [65] ^3^
SDAP supports and helps maintain the immune system during stressful events, such as:	
− weaning/separation	[42] ^3^, [66] ^3^, [67] ^3^;
− digestibility	[34] ^1^, [35] ^1^
− medical treatment	[68] ^2^; [69] ^3^;
− heat	[70] ^3^;
− training and travel	[31] ^3^
SDAP helps maintain normal digestion and intestinal health. SDAP helps maintain intestinal balance. SDAP improves gut health.	[34] ^1^, [49] ^1^, [58] ^2^, [71] ^2^, [52] ^2^, [53] ^2^, [72] ^2^, [73] ^2^, [74] ^2^, [75] ^2^, [54] ^3^, [55] ^3^, [56] ^3^;
SDAP reduces fecal output.	[34] ^1^
SDAP contains bioactive peptides and globulin proteins.	[76] ^2^, [77] ^2^, [78] ^2^, [79] ^2^, [80] ^2^, [81] ^2^, [82] ^2^.
SDAP helps maintain normal cognitive functions in senior animals.	[5] ^2^, [83] ^2^, [68] ^2^ [12] ^2^, [13] ^2^.
SDAP has prebiotic effects. SDAP develops beneficial intestinal microbiota.	[9] ^2^, [45] ^3^, [46] ^3^, [84] ^2^.

^1^ References about the target species (dogs and cats); ^2^ references about model mice and rats; ^3^ references about farm animals.

## Data Availability

Not applicable.

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
