# Peer review of "Spray-Dried Animal Plasma as a Multifaceted Ingredient in Pet Food"

_animals, 2023, doi:10.3390/ani13111773_

Round 1

Reviewer 1 Report (Previous Reviewer 2)

General Comments

The paper has been improved since the last submission and all of my previous major concerns have been addressed. Below are minor line-specific comments, mostly to improve clarity and reduce the number of run-on sentences.

My only major concern was regarding Table 1. The table also includes the FEDIAF recommendations for dogs and cats and this should be noted in the table title. Please include the units for the amino acids; is it a percentage of the feed ingredient or a percentage of the protein content? Also, please provide a reference for where the amino acid composition values in the table come from. In the text, (line 200-201), there is a reference to Table 1 and Purina’s Society Report and I am not sure that reference fits here. I could not find information in that reference about the main protein sources used in pet food. Please replace that reference in this sentence.

Line-Specific Comments

Line 33-37: This is a long sentence and could be split into two sentences for improved clarity. Perhaps something like, “The humanization desire for high-quality food for dogs and cats along with the demand for highly digestible, palatable, and biologically valuable ingredients has created challenges and opportunities. There is increased interest in protein ingredients that have functional properties, meet nutritional requirements, optimize health, and reduce the environmental impact.”

Line 93-95: This sentence could be improved. Perhaps, “Especially, meat co-products minimize environmental impacts and have been the focus of research to enhance their nutritional value [16,17].”

Line 109-112: Another long sentence. This sentence can be split after “…plasma concentration before drying. The drying stage uses reverse osmosis to remove water for the product with…”

Line 165: A couple of commas are needed here. “…other hydrocolloids, such as xanthan gum, guar gum, or locust bean gum, are the most…”

Line 190-192: This sentence is somewhat confusing. Are you dividing the amount of each amino acid in the feed ingredient by the recommended intake of that amino acid from the FEDIAF? If so, I recommend re-wording the sentence to be clearer.

Line 209: “deficient” should be changed to “low” unless the amount of methionine in SDAP is inadequate compared to the methionine requirement.

Line 231: “but” should be “and” as the second clause of the sentence agrees with statements made in the first clause of the sentence (higher DM digestibility = reduction in fecal volume).  

Line 320 – 325: This is a very long sentence. The authors should consider splitting this sentence into 2 or 3 sentences.

Line 363-365: Please provide a reference for this sentence.

Line 425-426: This sentence doesn’t really describe a limitation of SDAP, but rather a statement that introduces a limitation. The sentence could be altered to say “Third, researchers still need to investigate whether SDAP should be applied before or after extrusion for optimal results,” or something to that effect.

Some sentences are quite long and could be shortened to improve readability. There are also a couple of sentences that are confusing. These are noted in the above section under line-specific comments. 

Author Response

Reviewer 2 Report (Previous Reviewer 1)

Nice job

None

Author Response

Thanks for the suggestions, all of them have been accepted.

This manuscript is a resubmission of an earlier submission. The following is a list of the peer review reports and author responses from that submission.

Round 1

Reviewer 1 Report

Nice review, but please modify the following:

1. change the alignment of the text to "justify"

2. Please add a table of the known effects of SDAP in different animals species, and if possible in pets 

3. Please add the potential challenges/pitfalls of using SDAP

4. Please add future directions of SDAP

Author Response

Dear reviewer,

We would like to thanks for your consideration to our manuscript "Spray-Dried Animal Plasma as a Multifaceted Ingredient in Pet Food". Your contribution was very important to improve its quality. 

Unfortunately, as we need teo or three more weeks to make the revision as suggested by the reviewers, we decided to accept all the comments and also to make all the suggested modification, but we will perform the changes and after we will send the review as "a new submission". Thank you very much for your comments and we hope you can contribute again when we re-submit the paper.

Best regards

Reviewer 2 Report

This paper summarizes the benefits of spray-dried animal plasma (SDAP) for use in dog and cat food. The paper is well-organized, includes many descriptive tables and figures, and thoroughly discusses the benefits (or potential benefits) of SDAP for pets.

General comments

While this paper does a nice job describing the benefits of SDAP, there is no mention of any limitations of SDAP. I believe this review would be strengthened with some discussion of potential concerns or issues. These could include safety/pathogen control, ease of application to kibble (spray-coated versus incorporated into kibble), and cost.

I think it would benefit the paper to discuss typical inclusion rates of SDAP in dog and cat food and if these rates differ depending on food form (wet food versus kibble). This discussion may fit either in the introduction section or in the wet food section when talking about how many pet foods declare SDAP in their ingredients. From piecing together parts throughout this manuscript, it appears that inclusion rates in research studies range from 1.5 to 20%, which is quite broad! Is this indicative of the range seen in the industry?

In the conclusion, the authors make a point that inclusion rates differ in studies about nutrition versus functional properties. I think this is a very interesting and important point that should be introduced earlier on in the paper (see previous paragraph) and highlighted throughout the paper when discussing the studies.

There are a few grammatical errors throughout the paper (discussed below), but overall, these do not limit the readability of the paper.

Specific comments

Line 19: remove the word “induce” so that all the items in the list are consistent (‘digestibility’ = noun; ‘fecal quality’ = noun; ‘induce intestinal inflammation’ = verb but ‘intestinal inflammation’ = noun).

Line 24-26: This sentence may be better worded if “…it is discussed…” is removed and “…are discussed” is added at the end of the sentence

Figure 2: This figure lists characteristics of SDAP, but does not indicate whether the listed qualities are increased/positively affected or decreased/negatively affected with the addition of SDAP. For example, cognition may be improved, but inflammageing may be reduced. I recommend the addition of some sort of indication, such as up and down arrows, as to the effect SDAP has on each characteristic.

Additionally, there is a group of bacteria connected to the prebiotic bubble, which is slightly confusing to me. Does this indicate that SDAP is a prebiotic that feeds/increases the concentrations of these specific bacteria? Perhaps clarifying this relationship somehow would improve the interpretation of the figure.

Line 80: What is the “concentration membrane” used? Is this part of the processing of SDAP? If so, it would be good to include a description of concentration membranes in the introduction when discussing SDAP processing.

Line 82: Change “its” to “their” 

Line 218: “it is” should be “its” (no apostrophe)

Line 249-254: These two sentences explaining what AAS is should be moved up to the section that first introduces AAS (line 221-222)

Figure 6: The grey colors in this figure are difficult to distinguish. It would be easier to read if bars with different colors or textures were used.

Line 300-340: This section is mostly discussing immune properties and would fit better in Section 4 about Immunological effects of SDAP.

Table 2: It may be difficult to claim that a prebiotic effect was caused by immunological modifications. It is possible that the SDAP served as a prebiotic for beneficial bacteria which in turn affected the immune system. Additionally, the paper cited for this table does not mention prebiotics, so “prebiotic effect” will need a separate citation if it remains in this table. I recommend removing this claim from the table or providing a citation and a clearer description.

Line 358-363: This sentence is long and confusing. A prebiotic is a substance that feeds beneficial microbes in the gut. SDAP may indeed feed beneficial bacteria, but that is not made clear or supported well in this sentence. If a prebiotic claim is to be made, more evidence regarding SDAP’s effect on beneficial bacteria should be included. This sentence should also be divided into at least two sentences to improve clarity.

Line 382: remove “mainly”

Line 384: remove “and”

Line 392-394: this sentence should be reworded to avoid the use of “however” twice in the same sentence.

Line 438-443: This is another long sentence that should be divided into at least two sentences

Section 6. SDAP as a sustainable ingredient: I think this section may better fit earlier on in the manuscript, perhaps after the introduction which discusses how SDAP is produced.

Line 461-464: This sentence reads like an introductory sentence. It may fit better in the introduction or may be eliminated altogether, as all of the concepts in this sentence have already been discussed in other sections.

Author Response

(The authors gave the same response as above.)

Reviewer 3 Report

This is a well-written review of SDAP in pet food and will be well received. There are a few editorial comments that may improve the readability of the manuscript.

Line 32: change has to have (you describe plural forms of issues)

Line76. Include a brief list of oncoming sections. That will help the reader. For example: The following areas will be covered: wet pet food processing, nutritional value for dogs and cats, immunological effects, etc....

Line 80: Define "concentration membranes". This sentence won't make sense to most readers.

Line 94: change "its" to "the"

LIne 105: how is "juiciness" determined? Can you provide a scale or definition from the literature?

Line 148-164: Seems like this is a new section is in order as these paragraphs discuss health influences not nutritional aspects. 

Line 170: Provide the calculation here for AAS. That will help the reader, significantly.

Line 218: Change "it is" to "the"

Line 244: Remove "that" and change "resulted" to "resulting"

Line 300: This appears to be a new section on Immune Function. Seems like this section to line 340 should be included in the following section on Immunological effects.

LIne 414: change "parts" to "part results"

Line 495: It seems if you conclude prebiotic, immunomodulatory and neuroprotective properties, then that should be the outline? Definitely presented immunomodulatory effects, but not nearly the influence as a prebiotic or neuroprotective agent (these 2 don't seem supported by this review).

LIne 496: "Nowadays" is very lay terms. Change to "The technological benefits of SDAP..."

Author Response

(The authors gave the same response as above.)
